# Does local government tax competition promote industrial transformation and upgrading?

**Yuxin Meng[1], Xiaodong Yang[2], Chunji Zheng** [3]*

1 College of Finance and Taxation, Xinjiang University of Finance and Economics, Urumqi, China,
2 College of Economics and Management, Shihezi University, Shihezi, China, 3 College of Business,
Xinjiang Normal University, Urumqi, China

* zhengchunjilab@126.com

## Abstract

Promoting industrial transformation and upgrading (ITU) is the vital driving force for achieving high-quality economic development. This paper systematically interprets the theoretical mechanism of the impact of local government tax competition on ITU from a theoretical perspective. On this basis, a two-way fixed effects model is used to examine the impact of local government tax competition on ITU, the transmission mechanism, and the moderating effect of technological innovation. The study finds an inverted "U-shaped" relationship between local government tax competition and ITU, with tax structure, income distribution, and industrial agglomeration playing a non-linear mediating role. Improvements in technological innovation will reinforce the inverted "U-shaped" relationship between local government tax competition and ITU. The inverted "U-shaped" impact of local government tax competition on ITU is more pronounced in cities without economic growth targets, cities in eastern China, economically developed cities, and cities with weak tax enforcement capabilities. The research findings provide theoretical support and decision-making references for optimizing local tax policies and promoting high-quality economic development.

## 1. Introduction

Over the past four decades of reform and opening-up, China has achieved remark-able accomplishments in the economic sphere. However, due to long-standing weaknesses in foundational capabilities, a lack of core technologies, and low product value-added, China's industrial structure transformation has lagged. Many industries remain in the lower to middle segments of the global value chain, and the issue of being "Choked off" by key core technologies persists [1]. Therefore, China's current industrial system has yet to effectively break free from the constraints of being large but not strong and comprehensive but not excellent [2]. The report of the 20th Party Congress emphasizes the need to accelerate the development of a modernized industrial system. It emphasizes the need to expedite the adjustment

**Data availability statement:** The data is publicly accessible at the Dryad repository where readers may directly retrieve the relevant data for research verification and citation purposes: DOI: 10.5061/dryad.fj6q5747v.

**Funding:** The author(s) received no specific funding for this work.

**Competing interests:** The authors have declared that no competing interests exist.

and optimization of industrial structures. This requires that, in the process of establishing a new stage of development, implementing new development concepts, and constructing a new development pattern, we focus on industrial development strategic tasks, deeply promote industrial transformation and upgrading, and promote high-quality economic development [3]- [4]. Therefore, how to better facilitate industrial transformation and upgrading has become a crucial topic that requires in-depth study and resolution.

As an essential element of Chinese decentralization, local government tax competition impacts local industrial transformation and upgrading. Tax competition is a strategic game. Local governments, driven by dual incentives from the "Economic competition" and "Promotion competition", compete to introduce various attractive tax policies, such as tax incentives, rent reductions, and tax rebates, to attract mobile production factors [5]- [6]. Although the central government uniformly sets the tax rate in China, local governments can make specific adjustments to the local effective tax rate within the tax rate stipulated by law and the adjustable range. Whether it is the "Levy first and return later" in tax collection and management, the "Discretion" in the process of tax inspection, or the "Flexible operation" in the qualification of the main body of tax incentives, all of them will create regional differences in effective tax rates through the "Tax system". The formation of regionally differentiated effective tax rates will have a siphoning effect on the industrial capital of neighboring regions, realizing the reallocation of inter-regional resources and the spatial agglomeration of industries [7]. In addition, the "Promotion competition" among local officials serves as a political incentive for local governments to compete for tax revenue. In the promotional game of local officials, economic performance is a key factor in assessing their performance. To meet the promotion needs of local officials, they lower the effective tax rate of the jurisdiction to influence the industrial layout to achieve rapid economic growth [8]. This leads to the question addressed in this paper: Does the current increase in tax competition among local governments facilitate industrial transformation and upgrading? If the answer is yes, through what transmission mechanisms does local government tax competition influence industrial transformation and upgrading? Is it possible to design a mechanism to strengthen the role of local government tax competition in promoting industrial transformation and upgrading? Against the background of the deepening reform of the local fiscal and taxation system and the urgent need to upgrade the domestic industrial structure, an in-depth discussion of the above questions can not only provide new empirical evidence for the institutional design of tax competition but also be of great significance for realizing the high-quality development of China's economy in the new era.

The possible marginal contributions of this paper are reflected in the following three aspects: Firstly, compared with previous studies on the relationship between local government tax competition and industrial upgrading, this paper focuses on exploring the dynamic relationship between the two, characterized by a pattern of "First promoting, then inhibiting". At the same time, it examines the heterogeneous effects from multiple perspectives, including economic development goals, geographical location, city rankings, and the capabilities of tax officials. This provides

decision-making references for the differentiated implementation of fiscal and tax system reforms and the promotion of industrial upgrading. Secondly, unlike existing studies that only explain the mechanism of local government tax competition influencing industrial transformation and upgrading from the perspective of resource allocation, this paper analyses and tests the mechanism of local government tax competition influencing industrial transformation and upgrading from three aspects: tax structure, income distribution, and industrial agglomeration. This enriches the research on the transmission path of local government tax competition influencing industrial transformation and upgrading. Thirdly, it further incorporates technological innovation into the research framework and systematically examines the synergistic role played by technological innovation in local government tax competition affecting industrial transformation and upgrading, which expands the research field of local government tax competition and industrial transformation and upgrading. Fig 1 shows the research framework of this study.

## 2. Literature review

Industrial transformation and upgrading, characterized by multi-body and multi-level dynamics, is a strong driving force and core content of kinetic energy conversion and quality improvement in the new period [3]. Research in this area mainly focuses on two aspects. Firstly, systematically define the connotation of improving the effectiveness of industrial transformation and upgrading, and propose corresponding measurement methods. Previous literature has primarily focused on the two dimensions of industrial structure rationalization and industrial structure advancement to analyze the quantitative effects of industrial transformation and upgrading. The former focuses on reflecting the degree of coordination of the development of the three times industrial structures and the effective utilization of factors to measure the equilibrium

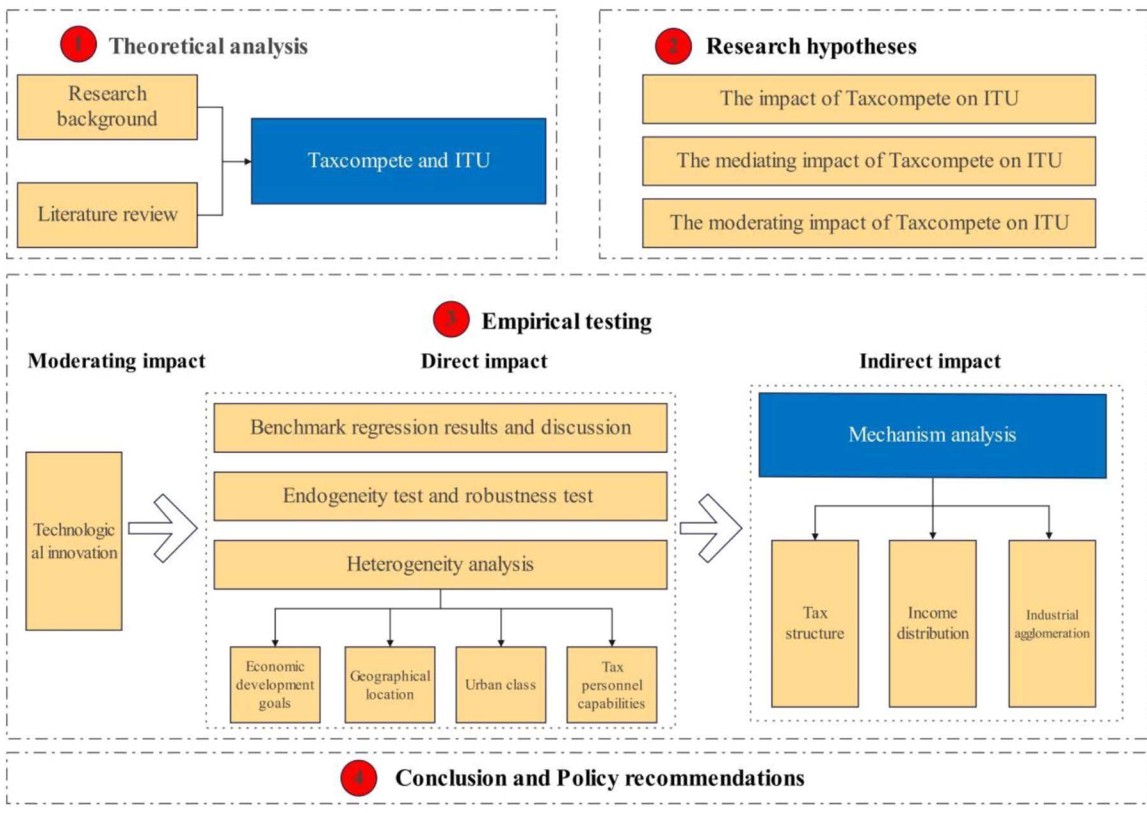

**Fig. 1. Research framework graph.**

characteristics of the evolution of industrial structure [9]- [10]; The latter refers to the evolution of industries from low to high levels. Scholars have conducted multidimensional, dynamic assessments and measurements from various aspects, including industrial structure advance coefficients, explicit comparative advantages of industries, and employment shares [11]- [12]. This effectively assesses the trends and patterns of industrial structure transformation from low-value-added to high-value-added industries from different perspectives. As a result, the above methodology has been widely applied by scholars to related research [13]. Secondly, focus on promoting industrial transformation and upgrading by conducting an in-depth exploration of the driving factors behind this process. Previous literature has examined this topic from various angles, including digital economic development [14], environmental regulations [15], local fiscal pressures [16], science and technology system reform, and innovation-driven strategies [17]. Combining theoretical analysis with empirical testing to examine their impact on industrial transformation and upgrading and the underlying mechanisms at play.

Academic studies related to local government tax competition can be roughly categorized into three types. Firstly, fiscal tax shock. Local government tax competition has led to reduced tax enforcement efforts by tax authorities, resulting in lower enforcement rates and increased corporate tax evasion [18]. There have even been cases of enterprises "Bargaining" with the Government in turn, resulting in a sharp decline in local tax revenues, especially from liquidity factors [19], and an increased reliance on non-tax revenues. This not only undermines the sustainability of local finances but also indirectly leads to an insufficient supply of public goods through tightened budget constraints [20] and reduces overall social welfare [21]. The second is the economic growth effect. Koethenbuerger and Lockwood (2010) [22] introduce stochastic shocks to productivity in an endogenous growth model, revealing how tax competition affects economic growth from a theoretical perspective. Some scholars have found that local tax competition in China significantly reduces FDI spillovers [23], corporate profitability [24], and growth performance [25], and significantly inhibits regional green development [26]. Some scholars are generally critical of tax competition among local governments. Under the ideology of economic supremacy, most recommendations suggest that the central Government should strengthen supervision and strictly regulate or restrict local governments' tax competition practices. A few scholars hold the opposite view, finding that local tax competition in China significantly promotes corporate exports [27] and technological innovation [28], albeit with slight differences in the specific tax types. Thirdly, the liquidity factor reset effect. Most local tax competition is targeted at the liquidity tax base, which directly affects the cross-regional mobility and spatial distribution patterns of production factors. It has been confirmed in the literature that local tax competition has a significant driving effect on the Cross-regional flow of capital [29], Business relocation, Cross-regional mergers and acquisitions [30], Subsidiary establishment [19]. The extent of the effect may be affected by the agglomeration economy, mobility costs, and central transfer payments.

Regarding the research on local government tax competition and industrial transformation and upgrading, through summarizing the existing scholars based on different research perspectives, research methods, and research data, it is found that at present, there are three main views of "Inhibition", "Promotion" and "Nonlinear" on the impact of local government tax competition on industrial transformation and upgrading. Researchers who support the "Inhibition theory" believe that tax competition will form incentive distortions, impede the establishment of a unified market, and guide the irrational flow of resources, which is not conducive to industrial transformation and upgrading [31]. Scholars who advocate the "Promotion theory" of tax competition believe that local government tax competition attracts high-technology industries to cluster in the region using competing to provide tax incentives and rent reductions, thus promoting the development of high-technology industries and helping to promote industrial transformation and upgrading [32]. Additionally, some scholars argue that the relationship between local government tax competition and industrial transformation and upgrading is nonlinear. For example, Tang and Ye (2020) [33] found that when there is a relative labor scarcity of capital in the jurisdiction, local government tax competition promotes industrial transformation and upgrading, while when there is a relative surplus of capital, the behavior of local government tax competition hinders the process of industrial transformation and upgrading.

Reviewing and combing through the relevant literature, studies on local government tax competition and industrial transformation and upgrading have been abundant. Scholars have also attempted to conduct a preliminary exploration

of the relationship between the two, providing a theoretical basis for this study. However, several aspects still require further expansion. Firstly, although academia has explored the relationship between local government tax competition and industrial transformation and upgrading, there is no consensus on the effects of local government tax competition on industrial transformation and upgrading. Furthermore, the heterogeneous impacts of local government tax competition on industrial transformation and upgrading under heterogeneous conditions have been overlooked. This may lead to biases in the assessment of the effects of local government tax competition. Secondly, there is a lack of in-depth exploration of the theoretical logic and mechanism behind the influence of local government tax competition on industrial transformation and upgrading. The existing literature primarily elaborates on the mechanism of local government tax competition affecting industrial transformation and upgrading from the perspective of resource allocation while overlooking the intermediary effects played by tax structure, income distribution inequality, and industrial agglomeration in the process. Therefore, there is still much room to explore the mechanism between the two. Finally, existing studies have neglected the synergistic effect of technological innovation on local government tax competition, which affects industrial transformation and upgrading. Technological innovation will promote industry development through improvements in labor productivity and the creation of new industries, facilitating industrial transformation and upgrading. Not only that but under the incentive of promotion mechanism, technological innovation has become an essential part of the promotion assessment of officials. To seize the first opportunity, local governments are more inclined to use tax incentives and various preferential subsidies for high-level talents, as well as other means, to strengthen their tax competitiveness. The purpose is to attract high-quality talent and draw high-tech enterprises, cultivate an information and technological advantage through innovation, accelerate collaborative innovation among enterprises, and ultimately lead to industrial transformation and upgrading.

## 3. Theoretical analysis and research hypothesis

### 3.1. The nonlinear impact of local government tax competition on industrial transformation and upgrading

In the early stage of local government tax competition, moderate tax competition can positively impact industrial transformation and upgrading. Firstly, tax competition prompts local governments to optimize the business environment and improve the efficiency of government services [34]. An efficient and clean government can provide businesses with better services and guarantees. It can attract high-quality businesses and capital with advanced technology and management experience, promoting local industrial transformation and upgrading. Secondly, the magnetic effect of local government tax competition provides the necessary material foundation for industrial transformation and upgrading. In the early stage of tax competition, local governments introduce tax incentives, which directly reduce the operating costs of enterprises and enhance their profit expectations [35]. This tax advantage is desirable to external investors seeking lower costs and higher profit margins, helping to drive the industry towards high-end, intelligent development and promoting industrial transformation and upgrading. Finally, under the pressure of tax competition, local governments will guide and support the development of high-tech, modern services and other high-end industries. At the same time, promoting the technological transformation and upgrading of traditional industries will help enhance local industries' competitiveness and sustainable development capabilities and promote industrial transformation and upgrading [36].

However, when the tax competition exceeds a specific limit, it may inhibit industrial transformation and upgrading, showing a downward trend in the second half of the inverted "U-shaped" relationship. Firstly, excessive tax competition reduces local government revenues and limits the funds available for public services [37]. Insufficient investment in public services such as education, medical care, and transportation will affect the city's living environment and business environment, which is not conducive to attracting high-end talents and enterprises and hinders industrial transformation and upgrading. Secondly, local governments may overly relax tax incentives to attract enterprises, leading to the entry of some low-quality and high-pollution enterprises, resulting in an irrational industrial structure [38]. At the same time, excessive competition may make local governments ignore local industrial foundations and advantages, unthinkingly follow the trend

to develop popular industries, triggering industrial isomorphism and vicious competition, destroying industrial ecology [39], and not conducive to industrial transformation and upgrading. Finally, excessive tax competition may make local governments focus too much on the short-term introduction of enterprises and tax revenue growth while neglecting the cultivation and support of the long-term innovation ability of enterprises [40]. Enterprises may lack the motivation for independent innovation due to over-reliance on tax incentives, affecting industrial transformation and upgrading. Based on this, this paper puts forward the following hypotheses:

**Hypothesis 1:** Local government tax competition and industrial transformation and upgrading are inverted "U-shaped" relationships.

### 3.2. Mediating effects of tax structure, income distribution and industrial agglomeration

At the initial stage of local government competition, to attract investment and enterprises, the government will often reduce the proportion of direct taxes directly linked to enterprise profits to reduce the tax cost of enterprises [41]. Enterprises can invest this saved capital in R&D, introduction of new technology and equipment, cultivation of high-end talents, and other vital aspects of enterprise development, which enhances their motivation and ability to carry out industrial transformation and upgrading. At the same time, the reduction in the proportion of direct taxes means that the burden of personal income tax and other burdens is reduced, and the actual income of laborers increases [42]. For high-quality talents, this improves their income expectations and quality of life, attracts their mobility to the region, provides intellectual support for industrial structure upgrading, and promotes the development of emerging industries and high-end service industries. In addition, industrial transformation and upgrading are often accompanied by higher-risk activities such as technological innovation and market development. After the reduction in the proportion of direct taxes, enterprises have sufficient funds to maintain operations when faced with innovation failures or market fluctuations. This reduces the risk of enterprises going bankrupt due to excessive tax burdens and encourages them to participate actively in innovation activities required for industrial upgrading [43]. It also promotes the development of the region's industrial structure towards high-end, intelligent, and green directions.

When the intensity of local government tax competition exceeds a reasonable threshold, it will lead to a sharp decline in fiscal revenue. In contrast, the direct tax base is relatively more stable and less susceptible to economic fluctuations. Increasing its share has become a rational choice for local governments to compensate for fiscal shortfalls [44]. However, the move by local governments to increase the proportion of direct taxes has directly squeezed corporate profit margins, weakening their ability to invest in research and development, equipment upgrades, and technological innovation. This has negatively impacted high-tech companies that rely heavily on research and development, thereby inhibiting industrial transformation and upgrading. At the same time, an increase in personal income tax reduces the actual income of innovative talent, weakens the region's attractiveness to them, and hinders the industry's transition to high-end and intelligent development. In addition, excessive tax competition may cause local governments to neglect infrastructure construction and public service provision to maintain fiscal balance, undermining the business environment necessary for industrial transformation and upgrading. Based on this, this paper puts forward the following hypotheses:

**Hypothesis 2:** Tax structure plays a mediating effect in the influence of local government tax competition on industrial transformation and upgrading.

In the early stages of local government tax competition, moderate tax incentives provided multidimensional impetus for industrial transformation and upgrading through optimizing income distribution. Local governments attract high-tech and high-value-added industries through tax breaks and preferential policies, creating many high-paying jobs with strict skill requirements. This directly raises the income level of residents, especially providing opportunities for upward mobility for low-income groups. At the same time, tax incentives for small and micro enterprises have significantly lowered the barriers to entrepreneurship and stimulated market vitality. They have opened up new avenues for employment and entrepreneurship for low- and middle-income groups, promoting a more balanced income distribution pattern [45]. The

equalization of income distribution directly promotes optimizing and upgrading residents' consumption structure. With the general increase in income levels, consumer demand for goods and services has shifted from necessities to higher quality, personalized, and high-end products. In response to these changes in market demand, companies have had to increase their investment in research and development, promote technological innovation, and enhance the technical content and added value of their products to accelerate industrial transformation and upgrading. At the same time, a more balanced income distribution has led to a diversified consumer market, with the consumption needs of different income groups being fully released [46]. This diverse demand has forced companies to expand their product ranges, develop personalized services, and drive the industry towards diversification and refinement, significantly enhancing the industry's overall competitiveness. In addition, income distribution improvements have profoundly impacted human capital accumulation [47]. As income distribution becomes more balanced, more families can afford education expenses, leading to a more equitable distribution of educational resources. This helps improve the overall quality of the workforce and provides a solid intellectual foundation for industrial transformation and upgrading. The continuous influx of high-quality talent can effectively promote enterprise innovation, accelerate the development of new technologies and business models, and thus inject lasting momentum into industrial transformation and upgrading.

However, when local government tax competition exceeds reasonable thresholds, it can lead to a decline in local fiscal capacity and worsen income distribution. On the one hand, the collapse of public services such as education, healthcare, and social security has left low-income groups unable to improve their skills due to educational and healthcare resources shortages. This has led to a human capital shortage in high-end industries, exacerbating class. It has also hindered transforming and upgrading technology-intensive industries such as intelligent manufacturing and information technology [48]. On the other hand, local governments may prioritize supporting high-energy-consuming, low-value-added industries that yield quick results to alleviate fiscal pressures. Such industries have stable tax bases and low tax administration costs, while strategic emerging industries requiring long-term investment receive insufficient support [49]. Social conflicts caused by income inequality may prompt local governments to prioritize maintaining stable employment, delaying the withdrawal of outdated production capacity and hindering industrial transformation and upgrading. Based on this, this paper proposes the following hypotheses:

**Hypothesis 3:** Income distribution gap plays a mediating effect in the influence of local government tax competition on industrial transformation and upgrading.

Local government tax competition has a linkage effect on the upstream and downstream of the industrial chain. On the one hand, local government tax competition in the early stages can attract investment from individual enterprises and more related enterprises to settle in through the synergistic effects of the upstream and downstream industries [50]. This industrial agglomeration phenomenon helps form a complete industrial chain, reduces enterprise transaction costs, and improves resource utilization efficiency. At the same time, industrial agglomeration can promote technological innovation and knowledge overflow, providing strong power support for industrial transformation and upgrading. On the other hand, the formation of industrial agglomeration can also play a synergistic effect and promote the overall transformation and upgrading of urban industries. The close cooperation and mutual support between different enterprises in the industrial chain can help form a good situation of complementary advantages and resource sharing [51]. This synergistic effect can further enhance urban industries' competitiveness and innovation ability and promote industrial transformation and upgrading.

Under excessive tax competition, local governments often use extraordinary tax incentives to attract businesses to compete for tax revenue. This leads to the disorderly clustering of industries in geographical space, ignoring regional carrying capacity and factor matching. This irrational concentration leads to excessive consumption of resources such as land, energy, and the environment, driving up production costs for enterprises and triggering vicious competition based on homogeneity. To survive, enterprises are forced to lower prices and cut back on R&D investment, which weakens the motivation for innovation [52]. When the scale of agglomeration exceeds the "Optimal density", the spillover effect

of knowledge diminishes. Talent and technology outflows intensify, high-end factors flee, and the innovation ecosystem breaks down. At the same time, local governments protect inefficient enterprises to maintain fiscal revenue and employment stability. This hinders the elimination of "Zombie enterprises" and prevents high-quality factors from flowing into emerging industries. Ultimately, the industrial agglomeration has shifted from promoting industrial transformation and upgrading as a "Positive externality" to inhibiting upgrading as a "Negative externality". This has created a vicious cycle of "Excessive competition—Agglomeration overload—Upgrading obstacles". Based on this, this paper proposes the following hypotheses:

**Hypothesis 4:** Industrial agglomeration plays a mediating effect in the influence of local government tax competition on industrial transformation and upgrading.

### 3.3. Moderating effects *of* technological innovation

The impact of local government tax competition on industrial transformation and upgrading is related to technological innovation. As the level of technological innovation improves, the extent of the effects of local government tax competition on industrial transformation and upgrading may also change. Improvements in technological innovation can, on the one hand, enhance corporate production efficiency and innovation conversion capabilities. This incentivizes companies to increase R&D investment and optimize production processes in a competitive tax environment, thereby directly promoting technological upgrades in the industry [53]. On the other hand, it can also promote the digital transformation of traditional industries and create new economic growth points through technological breakthroughs [54]. At the same time, we will use intelligent and information-based methods to transform traditional industries, thereby promoting industrial transformation and upgrading [55]. When local government tax competition does not exceed a reasonable threshold, improvements in technological innovation can optimize resource allocation efficiency and strengthen the agglomeration effect of innovation factors, thereby precisely converting the policy dividends brought about by tax competition into technological breakthroughs and industrial upgrading momentum [56]. This enhances the role of local government tax competition in promoting industrial transformation and upgrading. At a stage where excessive tax competition among local governments may be detrimental to industrial transformation and upgrading, technological innovation can help enterprises break their dependence on traditional competitive measures such as tax incentives. By developing new technologies and models, enterprises can improve production efficiency and reduce costs [57]. At the same time, technological innovation can also give rise to new industrial forms and business models. This injects new vitality into industrial transformation and upgrading, thereby alleviating the inhibitory effect of local government tax competition on industrial transformation and upgrading. Based on this, this paper puts forward the following hypotheses:

**Hypothesis 5:** Technological innovation positively regulates the inverted "U-shaped" relationship between local government tax competition and industrial transformation and upgrading.

## 4. Model setting, variable selection and data description

### 4.1. Modeling

To further empirically examine the impact of local government tax competition on industrial transformation and upgrading, this paper constructs the following benchmark regression model:

$$ITU_{it} = \alpha + \beta_1 Taxcompete_{it} + \beta_2 Taxcompete_{it}^2 + \gamma \sum_{k=1}^{6} X_{it} + \eta_i + \lambda_t + \varepsilon_{it} \tag{1}$$

In equation (1), $i$ denotes a prefecture-level city, $t$ denotes a year. $ITU_{it}$ denotes industrial transformation and upgrading; $Taxcompete_{it}$ denotes local government tax competition intensity; $Taxcompete_{it}^2$ denotes the squared term of local government tax competition intensity; $X_{it}$ denotes a series of control variables that may affect the industrial transformation and

upgrading, including the level of Economic Development (*Rgdp*), Financial capacity (*FC*), Infrastructure level (Road), Education level (Edu), the level of openness to the outside world (*Open*), and Population density (*PD*); $\beta$ is the core coefficient of most interest in this paper; $\eta_i$ is the Area-fixed effect, $\lambda_t$ is the Year-fixed effect; $\varepsilon_{it}$ is the random error term.

### 4.2. Selection of variables

**4.2.1. Explained variable: Industrial transformation and upgrading (*ITU_{it}*).** At this stage, industrial transformation and upgrading focus on economic benefits and social and environmental responsibilities. To measure industrial transformation and upgrading more comprehensively, this paper refers to the practice of Chen et al.(2022) [3], based on the new development concept, using the entropy method to construct a comprehensive evaluation system of industrial transformation and upgrading. Details of indicators are shown in Table 1.

**4.2.2. Explanatory variables: Local government tax competition (*Taxcompete_{it}*).** Local governments mainly compete for tax revenues by granting taxpayers different tax incentives, tax rebates, or reductions in non-tax revenues. The results of tax competition are directly reflected in changes in the relative tax burdens of local governments. This is because tax incentives mainly come from major tax types such as value-added tax and corporate income tax, and the tax sources of these tax types mainly come from the secondary and tertiary industries, with little involvement from the primary industry. Therefore, this paper draws on the research method of Chu et al. (2024) [58] to measure the relative tax burden of the region (*tax_{it}*) by the proportion of the tax revenue of the four main tax types, namely, VAT, business tax, enterprise income tax, and individual income tax, to the value-added of the secondary and tertiary industries. Then, the local government tax competition indexes (*Taxcompete_{it}*) are constructed based on the relative tax burden of each prefectural-level city from the level of neighboring regions to the national level. $atax_{min,it}$ represents the lowest relative tax burden among the adjacent areas of region i in year t, and $ttax_{mean,t}$ represents the national average relative tax burden in year t. The ratio between the two and the relative tax burden of this region is compared, and the larger the ratio, the lower the relative tax burden of this region and the greater the intensity of tax competition in this region.

$$Taxcompete_{it} = \frac{atax_{min,it}}{tax_{it}} \times \frac{ttax_{mean,t}}{tax_{it}} \tag{2}$$

**4.2.3. Control variables.** Considering the large number of factors affecting industrial transformation and upgrading, to avoid multicollinearity and endogeneity as much as possible, this paper adopts the following control variables to join the model to minimize the bias brought by these factors to the regression results. Among them, the Economic development level (Rgdp) is expressed by the logarithm of per capita GDP, which reflects the city's development

**Table 1. Evaluation system of industrial transformation and upgrading indicators.**

| Tier 1 indicators | Tier 2 indicators | Measurement methods | Indicator properties |
|---|---|---|---|
| Overall industrial structure | Advanced industrial structure | Spatial angle index | + |
| | Rationalization of industrial structure | Taylor index | − |
| Green development | Green productivity | Green total factor productivity | + |
| | Green production levels | Emissions of major pollutants per unit | − |
| Digital development | Digital industry level | Share of digital economy output | + |
| Technological and innovative development | Level of technological development | China Innovation and Entrepreneurship Regional Index | + |
| | Level of financial development | Peking University Digital Inclusive Finance | + |
| Urban-rural integration and development | Urban-rural dichotomy | Binary labor productivity comparison coefficient | + |
| | Level of agricultural mechanization | Total power of agricultural machinery/cultivated area | + |

potential and prompts the transformation of the industrial structure in the direction of high-end, intelligent, and green. Financial capacity (FC) is measured by the logarithm of the ratio of the general budgeted revenue to the general budgeted expenditures of the city's local finances. This indicator reflects the coordination capacity and sustainability of local government financial resources, which influence industrial transformation and upgrading by regulating the intensity of public service provision. Infrastructure level (Road) is measured by the logarithmic value of the ratio of paved road area to population in a city. This indicator reflects that well-developed transportation, communication, and energy infrastructure can significantly reduce the transmission time and cost of logistics, information flow, and energy flow, creating conditions for developing emerging industries and promoting industrial transformation and upgrading. Educational attainment (Edu) is measured by the logarithmic value of the ratio of the number of higher education teachers to the sum of the number of primary school teachers, secondary school teachers, and higher education teachers. This indicator provides high-quality talent, innovative thinking, and technical support for industrial transformation and upgrading. The level of openness to the outside world (Open) is measured by the ratio of total regional imports and exports to GDP. The higher this indicator, the more external resources can be introduced, the more markets can be expanded, and the more technological exchanges can be promoted, thereby facilitating industrial transformation and upgrading. Population Density (PD) is measured by the logarithmic value of the ratio of the region's total population to the jurisdiction's area at the end of the year, which provides sufficient human resources and consumer market support for industrial transformation and upgrading.

### 4.2.4. Mediating variables.

(1) Tax structure (TaxS) refers to the research method of Yu (2022) [59], using the logarithmic value of the ratio of direct tax to indirect tax to measure, in which the direct tax includes corporate income tax and personal income tax. The indirect tax includes value-added tax and business tax, and due to the policy of "Camp reform", the statistical data of the business tax is up to 2018.

(2) Income distribution (GAP) refers to the research method of Zhang and Chen (2018) [60] and is measured by the logarithmic value of the Taylor index. The larger the Taylor index represents, the higher the degree of income distribution inequality.

(3) The level of industrial agglomeration (IA) is based on Yang (2013) [61] research, using the ratio of the number of employed persons to the area of the administrative region to measure, the larger its value, the higher the level of industrial agglomeration.

### 4.2.5. Moderating variable.

Technological innovation (*Inno*) refers to the research method of Popp (2005) [62], using the logarithmic value of the total number of invention patents granted by the city to measure when the total number of invention patents is higher, it represents a higher level of technological innovation in the region.

### 4.3 Description of data

This paper uses data from 270 cities in China from 2011 to 2022 as the research sample. The study does not include Hong Kong, Macao, and Taiwan, with 3,240 observations. All original data for the variables in this paper are sourced from the EPS Global Statistical Database, Wind Database, China Statistical Yearbook, and China Urban Statistical Yearbook. Missing data were imputed using linear interpolation. Descriptive statistics for the variables are shown in Table 2.

## 5. Empirical results and analysis

### 5.1. Benchmark regression results

Table 3 reports the results of the benchmark regression on the impact of local government tax competition on industrial transformation and upgrading. To test the robustness of the model parameter estimation results, the estimation results are

**Table 2. Variable description statistics.**

| Variable | Obs | Mean | Std.Dev | Min | Max |
|---|---|---|---|---|---|
| ITU | 3240 | −3.0587 | 0.4856 | −4.2547 | −0.4228 |
| Taxcompete | 3240 | 0.6715 | 0.3226 | 0.1377 | 2.6110 |
| Rgdp | 3240 | 10.8260 | 0.6027 | 8.9713 | 22.9150 |
| FC | 3240 | 0.0183 | 1.3847 | −2.6564 | 9.3424 |
| Road | 3240 | 2.5034 | 0.5686 | 0.3922 | 5.0682 |
| Edu | 3240 | −4.7161 | 0.2860 | −6.6664 | −2.6497 |
| Open | 3240 | 0.1744 | 0.2660 | 0.0004 | 2.3735 |
| PD | 3240 | −3.4403 | 0.9240 | −8.2245 | −0.0890 |
| TaxS | 3240 | −1.0192 | 0.5415 | −3.4575 | 4.8897 |
| GAP | 3240 | −2.8391 | 0.6579 | −6.4410 | −1.3767 |
| IA | 3240 | 0.0119 | 0.0433 | 0.0001 | 0.9929 |
| Inno | 3240 | 5.1937 | 1.8235 | 0 | 11.3865 |

**Table 3. Benchmark regression results.**

| Variable | (1) | (2) | (3) | (4) | (5) | (6) | (7) |
|---|---|---|---|---|---|---|---|
| Taxcompete | 0.5253*** | 0.3858*** | 0.3882*** | 0.3728*** | 0.3154*** | 0.2038*** | 0.2307*** |
|  | (0.0806) | (0.0755) | (0.0755) | (0.0752) | (0.0730) | (0.0721) | (0.0693) |
| Taxcompete$^2$ | −0.1893*** | −0.1724*** | −0.1740*** | −0.1673*** | −0.1403*** | −0.1098$^2$ | −0.1157*** |
|  | (0.0427) | (0.0398) | (0.0398) | (0.0397) | (0.0385) | (0.0378) | (0.0363) |
| Rgdp |  | 0.2727*** | 0.2719*** | 0.2464*** | 0.2090*** | 0.1833*** | 0.1675*** |
|  |  | (0.0124) | (0.0125) | (0.0134) | (0.0132) | (0.0131) | (0.0126) |
| FC |  |  | −0.0109 | −0.0111* | −0.0107 | −0.0096 | −0.0143** |
|  |  |  | (0.0067) | (0.0067) | (0.0065) | (0.0064) | (0.0061) |
| Road |  |  |  | 0.0688*** | 0.0438*** | 0.0131 | 0.0271** |
|  |  |  |  | (0.0133) | (0.0130) | (0.0130) | (0.0126) |
| Edu |  |  |  |  | 0.3480*** | 0.2931*** | 0.2551*** |
|  |  |  |  |  | (0.0241) | (0.0240) | (0.0232) |
| Open |  |  |  |  |  | 0.3545*** | 0.2904*** |
|  |  |  |  |  |  | (0.0303) | (0.0294) |
| PD |  |  |  |  |  |  | 0.1534*** |
|  |  |  |  |  |  |  | (0.0094) |
| Cons | −3.3064*** | −6.1745*** | −6.1662*** | −6.0555*** | −3.9241*** | −3.8312*** | −3.3500*** |
|  | (0.0337) | (0.1346) | (0.1347) | (0.1359) | (0.1976) | (0.1937) | (0.1884) |
| Year-fixed effect | YES | YES | YES | YES | YES | YES | YES |
| Area-fixed effect | YES | YES | YES | YES | YES | YES | YES |
| R$^2$ | 0.4351 | 0.5087 | 0.5090 | 0.5129 | 0.5427 | 0.5613 | 0.5950 |
| N | 3240 | 3240 | 3240 | 3240 | 3240 | 3240 | 3240 |

Note: ***, **, * indicate significance at the 1%, 5% and 10% levels, respectively; () are error values. (Same below)

reported through the stepwise regression method by adding control variables one by one. From the results in Columns (1)-(7), it can be seen that the coefficients of the core explanatory variable Taxcompete are all significantly positive at the 1% level. The coefficient for Taxcompete 2 is significantly negative at the 1% level, and the coefficient has not changed substantially. Preliminary verification shows that local government tax competition has an inverted "U-shaped" relationship

with industrial transformation and upgrading (Linear part). However, at the same time, the growth rate will decrease by 0.1157 units (Non-linear part).

Furthermore, since the data may only exhibit a monotonically concave or curved relationship, it is necessary to conduct a U-test on the model containing the quadratic term for local government tax competition to prove whether there is a strict "U-shaped" relationship between the two. The results in Table 4 reject the original hypothesis, "There is no U-shaped relationship" at the 1% and above statistical level. The extreme point is 0.9969, within the range of [0.1377, 2.6110]. It can be seen from the "Slope value" that the slope of local government tax competition on the left side of the extreme point is positive and that on the right side of the slope is negative. The above results show that the inverted "U-shaped" relationship between local government tax competition and industrial transformation and upgrading exists significantly, which verifies hypothesis 1. Fig 2 shows that when the intensity of local government tax competition exceeds 0.9969, its adverse effect begins to appear. Industrial transformation and upgrading will decline with local government tax competition enhancement.

## 5.2. Endogeneity test

Local government tax competition and industrial transformation and upgrading may have a causal relationship with each other, causing endogeneity problems affecting the results' robustness. To minimize the impact of endogeneity on the regression results, drawing on the method of constructing instrumental variables by Chu et al. (2024) [58]. The cross-term between the lagged period of tax competition and the number of prefecture-level cities in the province is selected as the instrumental variable of tax competition, and the endogeneity test is conducted using the two-stage least squares method (IV-2SLS). As shown in Table 5, the coefficients of Taxcompete in (1)-(2) are both significantly positive at the 1% level. The

**Table 4. Utest test.**

| Variable | Lower bound | Upper bound |
|---|---|---|
| **Extreme point** | **0.9969** | |
| Interval | 0.1377 | 2.6110 |
| Slope value | 0.1988 | −0.3735 |
| t-value | 3.3079 | −2.9328 |
| P>|t| | 0.0005 | 0.0017 |
| Overall test | 2.93 | |

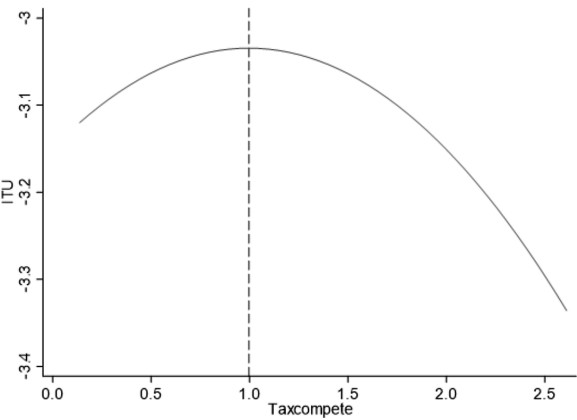

**Fig 2. The effect of Taxcompete on ITU.**

**Table 5. Endogeneity analysis.**

| Variable | (1) | (2) |
|---|---|---|
| First stage regression: | Taxcompete | |
| IV (Instrumental variable) | 0.0253*** | 0.0253*** |
| | (0.0011) | (0.0012) |
| Second stage regression: | ITU | |
| Taxcompete | 1.3305*** | 0.4300*** |
| | (0.2074) | (0.1423) |
| Taxcompete$^2$ | −0.5896*** | −0.2136*** |
| | (0.1004) | (0.0681) |
| Control variables | NO | YES |
| Year-fixed effect | YES | YES |
| Area-fixed effect | YES | YES |
| Kleibergen-Paap rk LM statistic | 209.522*** | 215.989*** |
| | [0.0000] | [0.0000] |
| Kleibergen-Paap rk Wald statistic | 492.850 | 468.571 |
| Cragg-Donald Wald F statistic | 1584.534 | 1453.259 |
| | {16.38} | {16.38} |
| R$^2$ | 0.4310 | 0.6055 |
| N | 2970 | 2970 |

Note: *** denotes significance at the 1% level; () is the error value, [] values are p-values, and {} values are critical values at the 10% level for the Stock-Yogo weak identification test.

coefficients of Taxcompete$^2$ are both significantly negative at the 1% level. This indicates a significant inverted "U-shaped" relationship between tax competition and industrial transformation and upgrading, consistent with the above regression results. In the first regression stage, the instrumental variables are significantly positive at the 1% statistical level, which means that the instrumental variables are positively correlated with local government tax competition and satisfy the correlation requirement. The Kleibergen-Paap rk LM statistic P-values are all 0.0000, firmly rejecting the null hypothesis of non-identifiability. Furthermore, the Kleibergen-Paap rk Wald statistic and Cragg-Donald Wald F statistic for the weak instrument variable test are greater than the standard value at the 10% level. Based on this, we conclude that the selected instrumental variables satisfy the correlation requirement, are strictly exogenous, do not suffer from weak instrumental variable issues, and are reasonably chosen. We further verify that the benchmark regression results of local government tax competition on industrial transformation and upgrading are robust.

### 5.3. Robustness tests

To ensure the reliability of the research conclusions on the impact of local government tax competition on industrial transformation and upgrading, robustness tests are required. In addition to adding control variables one by one, as used above, to test the robustness of the empirical results, a variety of robustness tests are conducted in this section, as shown in Table 6. (1) Replacement of core explanatory variables method. This section will redefine tax competition for robustness testing, drawing on Pu and Cheng's (2017) [63] measure using tax revenues as a share of gross local product, otherwise consistent with the above. The results are shown in columns (1)-(2). The coefficients of Taxcompete and Taxcompete$^2$ both passed the significance test at the 1% confidence level, indicating that the regression results mentioned above are robust. (2) Winsorizing. Winsorizing considers the possible outliers in the sample data that may interfere with the bench-mark results. This paper carries out a 1% tailing treatment on the sample data, and the results are shown in columns

**Table 6. Robustness testsl.**

| Variable | Replacement of core explanatory variables | | Winsorizing (1%) | | Core explanatory variables lagged one period | |
|---|---|---|---|---|---|---|
| | (1) | (2) | (3) | (4) | (5) | (6) |
| Taxcompete | 1.3e+03*** | 578.2971*** | 0.3126*** | 0.1358** | 0.5457*** | 0.2353*** |
| | (62.5008) | (61.6442) | (0.0759) | (0.0677) | (0.0856) | (0.0735) |
| Taxcompete$^2$ | −2.0e+05*** | −9.0e+04*** | −0.1106*** | −0.0729** | −0.1978*** | −0.1183*** |
| | (1.4e+04) | (1.3e+04) | (0.0400) | (0.0354) | (0.0453) | (0.0385) |
| Rgdp | | 0.1593*** | | 0.1558*** | | 0.1662*** |
| | | (0.0124) | | (0.0123) | | (0.0130) |
| FC | | −0.0162*** | | −0.0132** | | −0.0121* |
| | | (0.0061) | | (0.0060) | | (0.0064) |
| Road | | 0.0193 | | 0.0264** | | 0.0253* |
| | | (0.0124) | | (0.0123) | | (0.0129) |
| Edu | | 0.2204*** | | 0.2240*** | | 0.2635*** |
| | | (0.0232) | | (0.0227) | | (0.0243) |
| Open | | 0.2277*** | | 0.1777*** | | 0.3002*** |
| | | (0.0295) | | (0.0299) | | (0.0318) |
| PD | | 0.1455*** | | 0.1388*** | | 0.1489*** |
| | | (0.0093) | | (0.0092) | | (0.0097) |
| Cons | −3.4286*** | −3.4977*** | −3.2208*** | −3.3673*** | −3.3163*** | −3.3169*** |
| | (0.0189) | (0.1858) | (0.0317) | (0.1839) | (0.0358) | (0.1959) |
| Year-fixed effect | YES | YES | YES | YES | YES | YES |
| Area-fixed effect | YES | YES | YES | YES | YES | YES |
| R$^2$ | 0.4923 | 0.6047 | 0.3955 | 0.5301 | 0.4416 | 0.6002 |
| N | 3240 | 3240 | 3216 | 3240 | 2970 | 2970 |

(3)-(4). The coefficients of Taxcompete and Taxcompete$^2$ pass the significance test by at least a 5% confidence level, indicating that the above regression results are robust. (3) Core explanatory variables lagged one period. Considering the possible time lag effect of local government tax competition, this paper takes the lagged period of local government tax competition as the core explanatory variable for regression, and the results are shown in columns (5)-(6). The coefficients of Taxcompete and Taxcompete$^2$ both pass the significance test at a 1% confidence level, indicating that the regression results mentioned above are robust.

As shown in Table 7, (4) Replacement of sample range. In the study, to reduce the special economic characteristics of municipalities, policy preferences, resource allocation differences, extreme values, or outliers on the study results of possible interference and influence, this paper excludes the data of Beijing, Shanghai, Tianjin, Chongqing, and four municipalities. The results are shown in columns (1)-(2). The coefficients of Taxcompete and Taxcompete$^2$ both pass the significance test at a 5% confidence level, indicating that whether or not the municipality directly under the central government and separately listed cities are included does not affect the basic conclusions of this paper. (5) High-dimensional fixed effects. High-dimensional fixed effects can often eliminate the impact of many confounding factors. For this reason, in the benchmark regression based on the double fixed effects, increase the year and Area-fixed effects interaction of fixed effects. The results are shown in columns (3)-(4). The coefficients of Taxcompete and Taxcompete$^2$ are significant, at least at the confidence level of 5%, indicating that the regression results above are robust.

**Table 7. Robustness testsll.**

| Variable | Replacement of sample range | | High-dimensional fixed effects | |
|---|---|---|---|---|
| | (1) | (2) | (3) | (4) |
| Taxcompete | 0.3213*** | 0.1417** | 0.5546*** | 0.1687** |
| | (0.0754) | (0.0665) | (0.0878) | (0.0736) |
| Taxcompete$^2$ | −0.1135*** | −0.0771** | −0.1927*** | −0.1065*** |
| | (0.0398) | (0.0348) | (0.0470) | (0.0388) |
| Rgdp | | 0.1554*** | | 0.1902*** |
| | | (0.0120) | | (0.0129) |
| FC | | −0.0152*** | | −0.0153** |
| | | (0.0059) | | (0.0071) |
| Road | | 0.0354*** | | 0.0392*** |
| | | (0.0121) | | (0.0125) |
| Edu | | 0.2258*** | | 0.2451*** |
| | | (0.0221) | | (0.0229) |
| Open | | 0.1732*** | | 0.2771*** |
| | | (0.0292) | | (0.0291) |
| PD | | 0.1395*** | | 0.1523*** |
| | | (0.0090) | | (0.0092) |
| Cons | −3.2357*** | −3.3839*** | −3.3472*** | −3.6542*** |
| | (0.0315) | (0.1792) | (0.0363) | (0.1897) |
| Year-fixed effect | YES | YES | YES | YES |
| Area-fixed effect | YES | YES | YES | YES |
| Year*Area fixed effect | No | No | YES | YES |
| R$^2$ | 0.3104 | 0.4754 | 0.3265 | 0.5464 |
| N | 3148 | 3148 | 3180 | 3180 |

## 6. Further analysis

### 6.1. Mediation effect test

To verify that local government tax competition may promote industrial transformation and upgrading through three mechanisms of action: adjusting the tax structure, reducing the inequality of income distribution, and increasing the level of industrial agglomeration, this paper conducts a mediation effect test. Given the problems of overuse and endogeneity bias of the traditional stepwise method test of mediation effect, this section draws on the research method of Jiang (2022) [64] on the mediation effect test, focuses on the credibility of the causal relationship between the explanatory variables and the explanatory variables, and constructs the model as follows:

$$M_{it} = \alpha_0 + \alpha_1 \text{Taxcompete}_{it} + \alpha_2 \text{Taxcompete}_{it}^2 + \omega X_{it} + \eta_i + \lambda_t + \varepsilon_{it} \tag{3}$$

Eq. (3) shows that $M_{it}$ represents the mediating variables, including tax structure (TaxS), income distribution (GAP), and industrial agglomeration level (IA). The variables on the right side of the equal sign are consistent with the baseline model and will not be repeated.

Table 8 shows the results of testing the three local government tax competition mechanisms. Columns (1)-(2) are the test results of tax structure. The regression coefficient of Taxcompete is significantly negative at the 1% level. The regression coefficient of Taxcompete$^2$ is significantly positive. This indicates that local government tax competition has a

**Table 8. Mediation effect test.**

| Variable | TaxS | | GAP | | IA | |
|---|---|---|---|---|---|---|
| | (1) | (2) | (3) | (4) | (5) | (6) |
| Taxcompete | −0.5701*** | −0.5493*** | −0.9427*** | −0.6415*** | 0.0408*** | 0.0265*** |
| | (0.1072) | (0.1071) | (0.0971) | (0.0884) | (0.0091) | (0.0090) |
| Taxcompete$^2$ | 0.1027* | 0.0974* | 0.3018*** | 0.2394*** | −0.0168*** | −0.0127*** |
| | (0.0530) | (0.0528) | (0.0514) | (0.0463) | (0.0048) | (0.0047) |
| Rgdp | | 0.0311 | | −0.2876*** | | 0.0037** |
| | | (0.0210) | | (0.0161) | | (0.0016) |
| FC | | −0.0424*** | | −0.0027 | | 0.0008 |
| | | (0.0076) | | (0.0078) | | (0.0008) |
| Road | | 0.0088 | | 0.0180 | | 0.0023 |
| | | (0.0229) | | (0.0160) | | (0.0016) |
| Edu | | 0.1154*** | | −0.0754** | | 0.0132*** |
| | | (0.0395) | | (0.0296) | | (0.0030) |
| Open | | −0.1117 | | −0.3127*** | | 0.0173*** |
| | | (0.0713) | | (0.0375) | | (0.0038) |
| PD | | 0.0550 | | −0.1072*** | | 0.0066*** |
| | | (0.0372) | | (0.0120) | | (0.0012) |
| Cons | −0.6934*** | −0.3100 | −2.3736*** | −0.1420 | −0.0061 | 0.0370 |
| | (0.0466) | (0.3386) | (0.0406) | (0.2404) | (0.0038) | (0.0245) |
| Year-fixed effect | YES | YES | YES | YES | YES | YES |
| Area-fixed effect | YES | YES | YES | YES | YES | YES |
| R$^2$ | 0.6293 | 0.6345 | 0.5540 | 0.6410 | 0.1027 | 0.1407 |
| N | 3240 | 3240 | 3240 | 3240 | 3240 | 3240 |

significant "U-shaped" relationship with the tax structure. That is, the tax structure plays a non-linear mediating role in the relationship between local government tax competition and industrial transformation and upgrading, and hypothesis 2 is verified. In the early stages of local government tax competition, to attract investment and businesses, governments often reduce the proportion of direct taxes directly linked to corporate profits to reduce the tax burden on businesses. Companies can invest the savings in key areas of business development, such as research and development, introducing new technologies and equipment, and training high-end talent, thereby enhancing their motivation and ability to transform and upgrade their industries. When the intensity of local government tax competition exceeds a reasonable threshold, it will lead to a sharp decline in fiscal revenue. Local governments are forced to increase the proportion of direct taxes, which directly squeezes corporate profit margins and weakens their ability to invest in research and development and equipment upgrades, thereby inhibiting industrial transformation and upgrading.

Columns (3)-(4) show the results of the test of income distribution. The coefficient of Taxcompete on income distribution is significantly negative at the 1% level. The coefficient of Taxcompete$^2$ is significantly positive at the 1% level. A "U-shaped" relationship exists between local government tax competition and income distribution inequality, indicating that local government tax competition can promote industrial transformation and upgrading by reducing income distribution inequality, and hypothesis 3 is verified. In the early stages of local government tax competition, local governments attracted high-tech and high-value-added industries through tax breaks and preferential policies, thereby increasing employment opportunities, alleviating income inequality, and promoting industrial transformation and upgrading. However, when local government tax competition exceeds reasonable thresholds, it can lead to a decline in local fiscal capacity. This can cause the collapse of public services such as education, healthcare, and social security. Low-income groups

cannot improve their skills due to a shortage of educational and healthcare resources, leading to a human capital gap in high-end industries and exacerbating class. This hinders transforming and upgrading technology-intensive industries such as intelligent manufacturing and information technology.

Columns (5)-(6) are the test results of industrial agglomeration. The primary coefficient of local government tax competition on industrial agglomeration is significantly positive at the 1% level. The coefficient of Taxcompete on income distribution is significantly positive at the 1% level. The coefficient of Taxcompete$^2$ is significantly negative at the 1% level. Local government tax competition has a significant inverse "U-shaped" relationship with industrial agglomeration. Industrial agglomeration plays a non-linear mediating role in the relationship between local government tax competition and industrial transformation and upgrading, and hypothesis 4 is verified. A series of preferential tax policies will be introduced in the early stages of local government tax competition. These policies will attract investment from individual enterprises and more related enterprises to settle in the area through the synergistic effects of the upstream and downstream industries in the industrial chain. The continuous expansion of industrial agglomeration helps to form a complete industrial chain and industrial cluster. This reduces enterprise transaction costs, improves resource utilization efficiency, and provides strong momentum for industrial transformation and upgrading. Under excessive tax competition, local governments often use extraordinary tax incentives to attract businesses to compete for tax revenue. This leads to the disorderly clustering of industries in geographical space, ignoring regional carrying capacity and factor matching. This irrational concentration leads to excessive consumption of resources such as land, energy, and the environment, driving up production costs for enterprises and triggering vicious competition based on homogeneity. Enterprises are forced to cut back on R&D investment to survive, weakening their drive for innovation. Ultimately, the industrial agglomeration has shifted from promoting industrial transformation and upgrading as a "Positive externality" to inhibiting upgrading as a "Negative externality".

## 6.2. Analysis *of* moderating effects

To further test the moderating effect of technological innovation on the nonlinear relationship between local government tax competition and industrial transformation and upgrading, this paper constructs the model as follows:

$$Ind_{it} = \alpha_0 + \alpha_1 \text{Taxcompete}_{it} + \alpha_2 \text{Taxcompete}^2_{it} + \alpha_3 Inno_{it} + \alpha_4 Inno_{it} \times \text{Taxcompete}_{it}$$
$$+ \alpha_5 Inno_{it} \times \text{Taxcompete}^2_{it} + \omega X_{it} + \varphi_i + \delta_t + \varepsilon_{it} \tag{4}$$

As shown in equation (4), Inno$_{it}$ is technological innovation, and the other variables are consistent with the benchmark model and will not be repeated.

Table 9 shows that the coefficient of Taxcompete$^2$*inno is significantly negative. This indicates that technological innovation positively moderates the inverted "U-shaped" relationship between local government tax competition and industrial transformation and upgrading. The increase in technological innovation strengthens the promoting effect before the curve's inflection point and mitigates the inhibiting effect after the inflection point, assuming that hypothesis 5 is verified. Technological innovation has made the left side of the original inverted "U-shaped" curve steeper, the right side flatter, and shifted the inflection point to the right. To visually illustrate the moderating effect of technological innovation, we have plotted a diagram showing the non-linear relationship between technological innovation and local government tax competition and industrial transformation and upgrading (Fig 3). In this graph, "Inno" represents the trend of the moderating effect of technological innovation; "Low Inno" indicates the trend when technological innovation is at the "Mean minus one standard deviation"; "High Inno" indicates the trend when technological innovation is at the "Mean plus one standard deviation". Therefore, improvements in technological innovation levels can enhance a company's production efficiency and innovation conversion capabilities. This incentivizes companies to increase their R&D investment in a tax-competitive environment, directly driving industrial transformation and upgrading. During phases where excessive local government tax competition may hinder industrial upgrading and transformation, technological innovation can help enterprises reduce

**Table 9. Analysis of moderating effects.**

| Variable | ITU | |
|---|---|---|
| | **(1)** | **(2)** |
| Taxcompete | −0.3334[*] | −0.2801 |
| | (0.1827) | (0.1744) |
| Taxcompete$^2$ | 0.2429[**] | 0.2153[**] |
| | (0.1010) | (0.0964) |
| inno | 0.0835[***] | 0.0456[***] |
| | (0.0139) | (0.0136) |
| Taxcompete*inno | 0.1447[***] | 0.1094[***] |
| | (0.0347) | (0.0334) |
| Taxcompete$^2$*inno | −0.0737[***] | −0.0650[***] |
| | (0.0194) | (0.0186) |
| Rgdp | | 0.1035[***] |
| | | (0.0128) |
| FC | | −0.0146[**] |
| | | (0.0059) |
| Road | | −0.0088 |
| | | (0.0123) |
| Edu | | 0.1464[***] |
| | | (0.0233) |
| Open | | 0.2247[***] |
| | | (0.0289) |
| PD | | 0.1033[***] |
| | | (0.0096) |
| Cons | −3.6998[***] | −3.5144[***] |
| | (0.0731) | (0.1889) |
| Year-fixed effect | YES | YES |
| Area-fixed effect | YES | YES |
| R$^2$ | 0.5874 | 0.6268 |
| N | 3240 | 3240 |

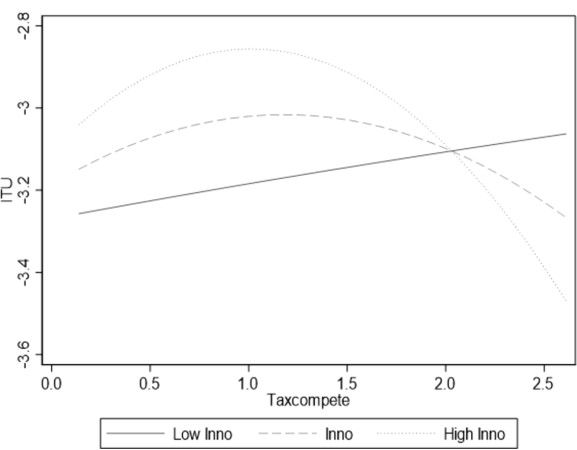

**Fig 3. Moderating effects of Inno.**

their reliance on traditional competitive measures such as tax incentives. It can also foster new industrial forms and business models, injecting new vitality into industrial upgrading and transformation, thereby alleviating the inhibitory effects of local government tax competition on industrial upgrading and transformation.

### 6.3. Heterogeneity analysis

(1) Economic development goals heterogeneity

To analyze whether local governments' tax competition strategies differ according to different economic development goals. This paper refers to Zhang et al. (2022) [65]. Firstly, we download the government work report for each city in the past few years on the official website of each city. Then, we use Python to write regular expressions to analyze the text of the government work report for each city in the past few years. Suppose specific and absolute expressions such as "To achieve a certain growth rate or more" exist. In that case, it is considered that the city's local government has set "Hard constraints" for the economic growth target of the following year. If there is an expression such as "Around" or a range to indicate the fluctuation range of the growth target, it is considered that the city's local government has set a "Soft constraint" for the following year's growth target. Accordingly, two dummy variables for "Hard constraints" and "Soft constraints" of local government economic growth targets are constructed. When a city meets the "Hard constraints" and "Soft constraints" of local government economic growth targets, it is set to 1; otherwise, it is set to 0.

The empirical results are shown in Table 10. When local governments set short-term economic growth targets as "Hard constraints", tax competition has a significant inverse "U-shaped" relationship with industrial transformation

**Table 10. Estimated results of Economic development goals heterogeneity.**

| Variable | Hard economic growth targets | | Soft economic growth target | |
|---|---|---|---|---|
| | Targeted | Untargeted | Targeted | Untargeted |
| Taxcompete | 0.2302*** | 0.2607* | 0.0817 | 0.2957*** |
| | (0.0798) | (0.1437) | (0.1098) | (0.0923) |
| Taxcompete$^2$ | −0.1143*** | −0.1619** | −0.0518 | −0.1534*** |
| | (0.0412) | (0.0789) | (0.0563) | (0.0492) |
| Rgdp | 0.1475*** | 0.2656*** | 0.1428*** | 0.1875*** |
| | (0.0139) | (0.0303) | (0.0180) | (0.0182) |
| FC | −0.0193*** | 0.0014 | −0.0096 | −0.0186** |
| | (0.0070) | (0.0129) | (0.0103) | (0.0080) |
| Road | 0.0427*** | −0.0261 | 0.0327* | 0.0190 |
| | (0.0149) | (0.0236) | (0.0194) | (0.0168) |
| Edu | 0.2400*** | 0.2819*** | 0.3273*** | 0.2197*** |
| | (0.0279) | (0.0419) | (0.0395) | (0.0291) |
| Open | 0.2814*** | 0.2758*** | 0.3608*** | 0.2188*** |
| | (0.0342) | (0.0581) | (0.0454) | (0.0395) |
| PD | 0.1409*** | 0.1783*** | 0.1631*** | 0.1441*** |
| | (0.0110) | (0.0180) | (0.0149) | (0.0123) |
| Cons | −3.2823*** | −4.0758*** | −2.6634*** | −3.7600*** |
| | (0.2154) | (0.4077) | (0.3050) | (0.2479) |
| Year-fixed effect | YES | YES | YES | YES |
| Area-fixed effect | YES | YES | YES | YES |
| R$^2$ | 0.5543 | 0.6926 | 0.6659 | 0.5277 |
| N | 2387 | 853 | 1287 | 1953 |

and upgrading. When local governments do not set challenging economic growth targets, tax competition still has a considerable inverse "U-shaped" relationship with industrial transformation and upgrading, but the impact is greater. When local governments set short-term economic growth targets as "Soft constraints", the coefficients of Taxcompete and Taxcompete² are insignificant. However, cities that do not put "Soft constraint" economic growth targets exhibit a significant inverted "U-shaped" relationship. This may be because when local governments set short-term economic growth targets as "Hard constraints", tax competition can, to a certain extent, attract enterprises to settle in the city, promote the increase of economic activities, and promote industrial development. However, as tax competition intensifies, it may excessively reduce tax revenues, leading to an insufficient supply of public services and lagging infrastructure construction, which is not conducive to industrial restructuring and upgrading, thus showing an inverted "U-shaped" relationship. The "Soft constraint" means that the enforcement of the target is relatively weak, and the local government's behavior in tax competition is unclear or lacks sufficient incentive to adjust tax policy to achieve specific economic growth targets actively. At this time, the impact of local government tax competition on industrial transformation and upgrading is vague, and it is difficult to show a clear linear or non-linear relationship. Without "Hard constraints" or "Soft constraints", local governments may be more flexible and autonomous in tax competition. They can focus on long-term industrial development planning and increase support for emerging and high-end manufacturing industries. At the same time, it also avoids the blind competition and resource waste that may occur due to the excessive pursuit of short-term economic growth, making tax competition's positive effect on industrial transformation and upgrading more significant. Similarly, an adverse impact causes an inverted "U-shaped" relationship to emerge after reaching a certain level.

(2) Geographical heterogeneity

As China is a vast country, the regression analysis based on the total sample of cities may hide the regional differences. This section divides China into eastern, central, and western regions according to geographic location to investigate the differential impact of local government tax competition on industrial transformation and upgrading. The empirical results are shown in Table 11. In the eastern region, local government tax competition has a significant inverted "U-shaped" relationship with industrial transformation and upgrading. In contrast, the inverted "U-shaped" relationship is insignificant in the central and western regions. This may be because the eastern region is more economically developed, with a relatively high-end industrial structure and a good foundation for industrial transformation and upgrading. Local government tax competition can also attract high-end industries and innovative elements to a certain extent, promoting the development of industries towards higher added value and high technology content. However, excessive tax competition among local governments may lead to a reduction in fiscal revenue. This could affect the provision of public services and infrastructure construction, negatively impacting industrial transformation and upgrading, resulting in an inverted "U-shaped" relationship. The economic development of central and western regions lags, with traditional industries dominating the industrial structure, making industrial transformation and upgrading difficult. The impact of tax competition on industrial transformation and upgrading is relatively small, and it is difficult to see a clear inverted "U-shaped" relationship. At the same time, the market size in central and western regions is relatively small, and resources are relatively limited. Local government tax competition plays a relatively weak role in attracting industries and resources, and has little effect on promoting industrial transformation and upgrading.

(3) Urban class heterogeneity

Different classes of cities have different resource accessibility, policy flexibility, and implementation strengths. This paper adopts the "2022 City Business Attractiveness Ranking" published by the New Tier 1 Cities Research Institute. It divides the sample cities into economically developed, relatively developed, and economically underdeveloped cities according to the first and second-tier cities, new tier 1 cities, and third, fourth, and fifth-tier cities, respectively. The

**Table 11. Estimated results of Geographical heterogeneity.**

| Variable | Eastern region | Central Region | Western Region |
|---|---|---|---|
| Taxcompete | 0.5624*** | 0.1697 | −0.0140 |
| | (0.1529) | (0.1349) | (0.1005) |
| Taxcompete$^2$ | −0.2894*** | −0.1462* | 0.0120 |
| | (0.0770) | (0.0838) | (0.0472) |
| Rgdp | 0.0945*** | 0.2588*** | 0.1784*** |
| | (0.0191) | (0.0243) | (0.0252) |
| FC | −0.0200* | −0.0435** | −0.0173** |
| | (0.0102) | (0.0199) | (0.0087) |
| Road | 0.0284 | 0.0051 | 0.0180 |
| | (0.0243) | (0.0200) | (0.0220) |
| Edu | 0.5917*** | 0.0989*** | 0.3548*** |
| | (0.0620) | (0.0273) | (0.0564) |
| Open | 0.2334*** | 0.1045 | 0.0458 |
| | (0.0464) | (0.0876) | (0.0569) |
| PD | 0.1896*** | 0.1422*** | 0.1089*** |
| | (0.0189) | (0.0175) | (0.0132) |
| Cons | −0.9314** | −5.0325*** | −3.0629*** |
| | (0.4064) | (0.3159) | (0.4117) |
| Year-fixed effect | YES | YES | YES |
| Area-fixed effect | YES | YES | YES |
| R$^2$ | 0.6484 | 0.3765 | 0.5104 |
| N | 1176 | 1176 | 888 |

empirical results are in columns (1)-(3) in Table 12. A significant inverse "U-shaped" relationship exists between local government tax competition in economically developed cities and industrial transformation and upgrading. The coefficient is lower in relatively developed cities. However, for economically underdeveloped cities, the coefficient of Taxcompete$^2$ is not significant and does not exhibit an inverse "U-shaped" relationship. This may be because economically developed cities have stronger economic strength and resources in the early stages of local government tax competition. By moderately reducing taxes and other measures, they can attract many high-quality enterprises, high-end talent, and innovative resources, promoting industrial transformation and upgrading. However, as local government tax competition intensifies, excessive tax cuts will significantly reduce fiscal revenue, which will affect investment in scientific and technological innovation and be detrimental to industrial transformation and upgrading. Compared with economically developed cities, relatively developed cities may have relatively moderate tax competition, as their economic strength and resources are less robust than those of developed cities. Local governments in these regions may engage in less intense tax competition and focus more on improving the overall business environment and industrial support systems. This attracts businesses and investment rather than relying solely on local government tax competition. Economically underdeveloped cities have weak economic foundations and limited resources, and local governments have relatively little room and capacity for tax competition. Even if they engage in tax competition, attracting sufficient high-quality resources to promote industrial transformation and upgrading is difficult. In addition, these cities may focus more on the scale and speed of economic growth rather than the quality of industrial transformation and upgrading. They do not attach enough importance to the role of local government tax competition in industrial transformation and upgrading, so the impact of tax competition on industrial transformation and upgrading is challenging to see.

**Table 12. Urban class and Tax personnel competencies heterogeneity.**

| Variable | (1) | (2) | (3) | (4) | (5) |
|---|---|---|---|---|---|
| Taxcompete | 1.2880*** | 0.2114* | 0.1734* | −0.0075 | 0.2977*** |
|  | (0.3702) | (0.1171) | (0.0887) | (0.1247) | (0.0862) |
| Taxcompete$^2$ | −0.6078*** | −0.1167* | −0.0633 | 0.0136 | −0.1761*** |
|  | (0.2044) | (0.0606) | (0.0484) | (0.0584) | (0.0490) |
| Rgdp | 0.5779*** | 0.1071*** | 0.0943*** | 0.1520*** | 0.1636*** |
|  | (0.0740) | (0.0148) | (0.0210) | (0.0275) | (0.0143) |
| FC | 0.0266 | −0.0236*** | −0.0050 | −0.0171* | −0.0217*** |
|  | (0.0211) | (0.0084) | (0.0093) | (0.0096) | (0.0080) |
| Road | −0.8437*** | 0.0210 | 0.0133 | 0.1402*** | −0.0077 |
|  | (0.0855) | (0.0187) | (0.0168) | (0.0250) | (0.0144) |
| Edu | 1.1205*** | 0.1357*** | 0.0588 | 0.1219*** | 0.3890*** |
|  | (0.1641) | (0.0343) | (0.0363) | (0.0313) | (0.0334) |
| Open | 0.6931*** | 0.2193*** | 0.0560 | 0.2125*** | 0.2759*** |
|  | (0.0877) | (0.0419) | (0.0519) | (0.0773) | (0.0329) |
| PD | 0.1670*** | 0.1250*** | 0.0751*** | 0.1960*** | 0.1449*** |
|  | (0.0558) | (0.0186) | (0.0124) | (0.0239) | (0.0102) |
| Cons | −2.1026* | −3.3039*** | −3.7871*** | −3.8466*** | −2.6462*** |
|  | (1.1079) | (0.2414) | (0.2946) | (0.3524) | (0.2429) |
| Year-fixed effect | YES | YES | YES | YES | YES |
| Area-fixed effect | YES | YES | YES | YES | YES |
| R$^2$ | 0.8736 | 0.4710 | 0.3132 | 0.6284 | 0.5883 |
| N | 216 | 1176 | 1848 | 956 | 2284 |

(4) Tax personnel capabilities heterogeneity

In the case of different abilities of tax personnel, there are differences in the effectiveness of tax policy implementation and the efficiency of tax collection and administration, which may affect the effect of local government tax competition on industrial transformation and upgrading. In this paper, referring to the study of Zhao and Sun (2023) [66], the median of the proportion of CPAs and CTAs among provincial tax personnel is taken as the cut-off point. When the proportion of the province in which a particular city is located is higher than the median, it is delineated as a city with high tax personnel capacity. Otherwise, it is a city with low tax personnel capacity. The data was manually compiled from the China Tax Audit Yearbook. The empirical results are in columns (4)-(5) in Table 12. The coefficients for Taxcompete and Taxcompete$^2$ are insignificant in cities with strong tax personnel capabilities. In cities with weak tax personnel capabilities, local government tax competition has a significant inverse "U-shaped" relationship with industrial transformation and upgrading. This may be due to the strong learning ability of tax officials to understand and implement tax policies better, which makes tax policies more standardized and stable. In this case, local governments might be less motivated to compete on taxes because they know that unfair tax competition could lead to more risks and problems. Tax officials with strong learning abilities often emphasize providing high-quality services to businesses. They enhance the attractiveness of their regions by improving the efficiency of tax services and providing professional tax consulting. This shift towards service orientation has led local governments to focus more on improving the overall business environment rather than simply competing on taxes. As a result, the coefficient of tax competition among local governments is insignificant. Cities with weak tax officials' learning capabilities may lack sufficient understanding and implementation of tax policies, leading them to rely more on traditional tax competition measures to attract businesses. Attracting investment through measures such as reducing tax rates and

offering tax incentives may have a particular promotional effect on industrial upgrading in the short term. However, as competition intensifies, excessive competition may arise, negatively impacting industrial transformation and upgrading. In such cases, local government tax competition is often short-term and blind, failing to consider the long-term impact of tax competition on industrial upgrading.

## 7. Conclusions and policy implications

Based on the panel data of 270 prefecture-level and above cities in China from 2011 to 2022, this paper applies the two-way fixed-effects model to test the role of local government tax competition on industrial transformation and upgrading. It draws the following conclusions: There is an inverted "U-shaped" relationship between local government tax competition and ITU, with tax structure, income distribution, and industrial agglomeration playing a non-linear mediating role. Improvements in technological innovation will reinforce the inverted "U-shaped" relationship between local government tax competition and ITU. The inverted "U-shaped" impact of local government tax competition on ITU is more pronounced in cities without economic growth targets, cities in eastern China, economically developed cities, and cities with weak tax enforcement capabilities. Based on theoretical analysis and empirical results, this paper proposes the following policy implications:

(1) Scientifically regulate the intensity of tax competition and avoid the "Excessive competition" trap. Local governments should establish a mechanism for assessing the reasonable range of tax competition. They should organize professional teams to conduct in-depth research, evaluate tax competition in their local areas and surrounding regions, and formulate differentiated thresholds for preferential tax policies. The central government should establish a filing and review system for preferential tax policies and intervene in local policies that exceed the reasonable range to prevent the "Restraining effect" of tax competition on industrial transformation and upgrading from manifesting too early.

(2) Increase support for technological innovation and optimize the innovation ecosystem. Local governments should increase financial investment in technological innovation and establish special innovation funds to support enterprises in their research and development activities. At the same time, the government can provide a favorable environment for industrial transformation and upgrade by investing in scientific and technological innovation infrastructure.

(3) Deepen tax structure reform and leverage intermediary roles. Firstly, an early warning system for tax competition thresholds must be established. Dynamically monitor changes in the proportion of direct taxes, set reasonable red lines for tax competition intensity, and avoid excessive reductions in direct taxes that could lead to fiscal imbalances. Secondly, the linkage between income distribution and public services should be strengthened to break the transmission chain of class. Implement a "Public service performance-linked" system, tying local government tax competition policies to metrics such as skill improvement and employment rates for low-income groups. Thirdly, guide industrial agglomeration toward a "Quality-oriented" transformation. Establish a "Supply chain tax incentive package" to provide joint R&D expense deductions for clusters of enterprises within agglomeration zones that form upstream technical collaborations.

(4) Formulate differentiated tax policies and strengthen industrial support. Cities without economic growth targets can focus more on the quality and sustainability of industrial transformation and upgrading, avoiding the blind pursuit of economic growth rates. Cities in eastern regions and economically developed cities can leverage their economic advantages and innovative resources to increase support for high-end and emerging industries. For cities with weak tax administration capabilities, strengthen training for tax officials to improve tax collection, administration standards, and service quality. For cities with relatively lagging economic development, the central government can provide support through fiscal transfers and other means to facilitate industrial upgrading and infrastructure development. It also encourages industrial cooperation and targeted assistance between developed and underdeveloped regions,

establishes special funds for industrial upgrading, and supports key industries and projects in underdeveloped areas. Encourage enterprises from developed regions to invest and establish factories in underdeveloped regions to drive local industrial development.

Although this study systematically examines local government tax competition's impact and transmission mechanism on industrial transformation and upgrading, some limitations remain. Only three mediating effects of tax structure, income distribution, and industrial agglomeration level are considered in this paper, and other mediating effects of local tax competition deserve further exploration.

## Supporting information

**S1 File. Modified data.**
(XLSX)

## Author contributions

**Conceptualization:** Xiaodong Yang.

**Data curation:** Yuxin Meng.

**Formal analysis:** Yuxin Meng.

**Funding acquisition:** Yuxin Meng.

**Investigation:** Chunji Zheng.

**Methodology:** Chunji Zheng.

**Project administration:** Chunji Zheng.

**Resources:** Chunji Zheng.

**Software:** Xiaodong Yang.

**Supervision:** Xiaodong Yang.

**Validation:** Xiaodong Yang.

**Visualization:** Yuxin Meng, Xiaodong Yang.

**Writing – original draft:** Yuxin Meng.

**Writing – review & editing:** Chunji Zheng.

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
