## [Decision Letter · Decision Letter 0]

21 Jul 2025

PONE-D-25-34958Does local government tax competition promote industrial transformation and upgrading?PLOS ONE?

Dear Dr. Zheng,

Thank you for submitting your manuscript to PLOS ONE. After careful consideration, we feel that it has merit but does not fully meet PLOS ONE’s publication criteria as it currently stands. Therefore, we invite you to submit a revised version of the manuscript that addresses the points raised during the review process.

We look forward to receiving your revised manuscript.

Kind regards,

Mingjun Hu

Academic Editor

PLOS ONE

Journal Requirements:

**Additional Editor Comments:**

Dear Dr. Zheng,

Thank you for submitting your manuscript to PLOS ONE. We have now received and reviewed the reports from two expert reviewers. After carefully considering their comments and the content of your manuscript, we would like to invite you to revise and resubmit your paper.

Both reviewers acknowledge the potential academic value of your work, particularly in addressing the nonlinear impact and transmission mechanism of industrial transformation under the framework of local tax competition, as well as the moderating role of technological innovation. However, they also raised several substantive concerns that must be addressed before the manuscript can be considered further.

The main issues identified include:

1.The manuscript requires substantial language polishing and refinement to improve clarity and avoid redundancy.

2.The literature review should be more concise and focused.

3.Greater consistency is needed in the theoretical logic, particularly in the discussion of direct taxation and tax competition.

4.The description and calculation of control variables should be supported by appropriate references.

5.The interpretation of regression results could be clearer, especially regarding the magnitude and meaning of coefficients.

6.It is recommended that English references be used wherever possible.

7.Some figure elements (e.g., variable definitions in Figure 3) should be better explained, ideally in footnotes.

In light of the above, we are inviting a major revision of your manuscript. We encourage you to carefully address all reviewer comments in your revision and provide a detailed response explaining how each point has been handled.

We appreciate your contribution to the journal and look forward to receiving your revised submission.

Best regards,

Mingjun Hu

Academic Editor

PLOS ONE

Reviewers' comments:

Reviewer's Responses to Questions

**Comments to the Author**

1. Is the manuscript technically sound, and do the data support the conclusions?

Reviewer #1: Yes

Reviewer #2: Yes

2. Has the statistical analysis been performed appropriately and rigorously?

Reviewer #1: Yes

Reviewer #2: Yes

3. Have the authors made all data underlying the findings in their manuscript fully available?

Reviewer #1: Yes

Reviewer #2: Yes

4. Is the manuscript presented in an intelligible fashion and written in standard English?

Reviewer #1: Yes

Reviewer #2: No

Reviewer #1: This article systematically explains the nonlinear impact and transmission mechanism of industrial transformation and upgrading from the perspective of local tax competition, and tests the regulatory effect of technological innovation between the two, which has very important research value and significance for promoting high-quality economic development. However, it is undeniable that the thesis still has room for improvement, and there are some flaws and deficiencies that need to be further supplemented and improved:

1.The second paragraph of Section 3.2 describes the negative impact on industrial transformation and upgrading when the intensity of local government tax competition exceeds a reasonable threshold. Local governments should further reduce the proportion of direct taxes when the intensity of tax competition exceeds a reasonable threshold. However, the sentence "however, the move by local governments to increase the proportion of direct taxes has directly squeezed corporate profit margins, weakening their ability to invest in research and development, equipment upgrades, and technical innovation." in the fifth line of this paragraph indicates that local governments should increase the proportion of direct taxes, which is contrary to the previous view.

2.Grammar needs further polishing.

3.It is also suggested that the calculation of control variables should also include references to relevant literature.

4.In the "Benchmark regression results", for the linear part, if the level of local government tax competition increases by 1 unit, how many more units will industrial transformation and upgrading increase?

5.It is recommended to use English references instead of Chinese references.

6.It is suggested that the explanations for "lnno", "Low lnno" and "High lnno" in Figure 3 be placed in the footnotes.

Reviewer #2: 1. The text in the article could be further refined.

2. The language could be further refined.

3. It is recommended that all references be in English.

4. References can also be included in the description of controlled variables.

5. The literature review could be more concise.

6. There are several repetitive expressions in the writing of the article. It is suggested that the author further simplify the language.

**Do you want your identity to be public for this peer review?** For information about this choice, including consent withdrawal, please see our Privacy Policy

Reviewer #1: No

Reviewer #2: No

---

## [Author Response · Author response to Decision Letter 1]

11 Sep 2025

Reviewers1:

Comments

This article systematically explains the nonlinear impact and transmission mechanism of industrial transformation and upgrading from the perspective of local tax competition, and tests the regulatory effect of technological innovation between the two, which has very important research value and significance for promoting high-quality economic development. However, it is undeniable that the thesis still has room for improvement, and there are some flaws and deficiencies that need to be further supplemented and improved:

(1) The second paragraph of Section 3.2 describes the negative impact on industrial transformation and upgrading when the intensity of local government tax competition exceeds a reasonable threshold. Local governments should further reduce the proportion of direct taxes when the intensity of tax competition exceeds a reasonable threshold. However, the sentence "however, the move by local governments to increase the proportion of direct taxes has directly squeezed corporate profit margins, weakening their ability to invest in research and development, equipment upgrades, and technical innovation." in the fifth line of this paragraph indicates that local governments should increase the proportion of direct taxes, which is contrary to the previous view.

R: We are most grateful for the reviewer's valuable comments. We fully acknowledge the importance of theoretical consistency for the rigour of the paper. To this end, we have refined the section on the mechanisms of direct taxation and tax competition, significantly enhancing the theoretical coherence of the relevant discussion. (p.6-p.8)

(2) Grammar needs further polishing.

R: We are most grateful for the reviewer's valuable comments. We fully acknowledge that the manuscript requires refinement in its linguistic expression. To this end, we have engaged a native English-speaking professional peer to conduct a systematic linguistic polish of the entire text. We have systematically reviewed each paragraph to identify and eliminate redundant content, thereby ensuring compliance with the linguistic standards of academic journals.

(3) It is also suggested that the calculation of control variables should also include references to relevant literature.

R: We are most grateful for the reviewer's valuable comments. We fully appreciate that in academic research, providing references for the description and computational methods of control variables enhances the rigour and reproducibility of the study, while also demonstrating the scholarly rationale for methodological choices. Accordingly, we have supplemented each control variable mentioned in the text with the relevant references. (p.13-p.14)

(4) In the "Benchmark regression results", for the linear part, if the level of local government tax competition increases by 1 unit, how many more units will industrial transformation and upgrading increase?.

R: We are most grateful for the reviewer's valuable comments. We fully concur that the clarity of regression result interpretations is crucial for enhancing the readability of research. To this end, we have supplemented the regression coefficients for the core explanatory variables with specific quantitative interpretations, particularly regarding the magnitude and meaning of the coefficients. (p.15)

(5) It is recommended to use English references instead of Chinese references.

R: We are most grateful for the reviewer's valuable suggestions. We fully concur that prioritizing English-language references in international academic publications constitutes a crucial principle for enhancing research traceability and ensuring accessibility for an international readership. This recommendation is accorded the utmost importance. The primary reason for retaining this non-English reference in our study lies in the unique comprehensiveness of its proposed methodology for measuring industrial transformation and upgrading. This approach serves as an indispensable cornerstone supporting our research's computational logic, rendering it irreplaceable. To adequately balance the necessity of the citation with the readability requirements of an international journal, we have provided a detailed explanation of the specific measurement methodology for industrial transformation and upgrading from the non-English source within the original text. This ensures international readers can accurately grasp its core concepts and role within this study. (p.12)

(6)It is suggested that the explanations for "lnno", "Low lnno" and "High lnno" in Figure 3 be placed in the footnotes.

R: We are most grateful for the reviewer's valuable comments. We fully appreciate the importance of clear explanations for chart elements in enhancing data readability, and acknowledge that certain figures in the original manuscript (such as Figure 3) lacked sufficient detail in their explanations regarding variable definitions, which may have hindered readers' accurate understanding of the results. Accordingly, we have added detailed explanations for all variables featured in Figure 3 within the caption text, ensuring their conceptual clarity. (p.23)

Reviewers2:

(1) The text in the article could be further refined.

R: We are most grateful for the reviewer's valuable comments. We fully acknowledge that the manuscript requires refinement in its Verbal expression. To this end, we have engaged a native English-speaking professional peer to conduct a systematic linguistic polish of the entire text. We have systematically reviewed each paragraph to identify and eliminate redundant content, thereby ensuring compliance with the linguistic standards of academic journals.

(2) The language could be further refined.

R: We are most grateful for the reviewer's valuable suggestions. We have streamlined and refocused the language throughout the text to more clearly underpin the research value of this study. (p.4-p.6)

(3) It is recommended that all references be in English.

R: We are most grateful for the reviewer's valuable suggestions. The primary reason for retaining this non-English reference in our study lies in the unique comprehensiveness of its proposed methodology for measuring industrial transformation and upgrading. This approach serves as an indispensable cornerstone supporting our research's computational logic, rendering it irreplaceable. To adequately balance the necessity of the citation with the readability requirements of an international journal, we have provided a detailed explanation of the specific measurement methodology for industrial transformation and upgrading from the non-English source within the original text. (p.12)

(4) References can also be included in the description of controlled variables.

R: We are most grateful for the reviewer's valuable comments. We fully appreciate that in academic research, providing references for the description and computational methods of control variables enhances the rigour and reproducibility of the study, while also demonstrating the scholarly rationale for methodological choices. Accordingly, we have supplemented each control variable mentioned in the text with the relevant references. (p.13-p.14)

(5) The literature review should be more concise and focused.

R: We are most grateful for the reviewer's valuable suggestions. We fully appreciate the importance of conciseness and focus in the literature review for enhancing the logical coherence of the paper. Accordingly, we have streamlined and refocused this section to more clearly underpin the research value of this study. (p.4-p.6)

6) There are several repetitive expressions in the writing of the article. It is suggested that the author further simplify the language.

R: We are most grateful for the reviewer's valuable comments. We have reviewed the entire text, removed redundant expressions, and further simplified the language.

---

## [Decision Letter · Decision Letter 1]

20 Oct 2025

Does local government tax competition promote industrial transformation and upgrading?

PONE-D-25-34958R1

Dear Dr. Zheng,

We’re pleased to inform you that your manuscript has been judged scientifically suitable for publication and will be formally accepted for publication once it meets all outstanding technical requirements.

Kind regards,

Mingjun Hu

Academic Editor

PLOS ONE

Additional Editor Comments :

Reviewers' comments:

Reviewer's Responses to Questions

**Comments to the Author**

Reviewer #1: (No Response)

Reviewer #2: All comments have been addressed

2. Is the manuscript technically sound, and do the data support the conclusions?

Reviewer #1: (No Response)

Reviewer #2: Yes

3. Has the statistical analysis been performed appropriately and rigorously?

Reviewer #1: (No Response)

Reviewer #2: Yes

4. Have the authors made all data underlying the findings in their manuscript fully available?

Reviewer #1: (No Response)

Reviewer #2: Yes

5. Is the manuscript presented in an intelligible fashion and written in standard English?

Reviewer #1: (No Response)

Reviewer #2: Yes

Reviewer #1: (No Response)

Reviewer #2: The authors responded well to the revision suggestions and the revised manuscript has been improved. I agree to publish the article.

**Do you want your identity to be public for this peer review?** For information about this choice, including consent withdrawal, please see our Privacy Policy

Reviewer #1: No

Reviewer #2: No

---

## [Editor Report · Acceptance letter]

PONE-D-25-34958R1

PLOS One

Dear Dr. Zheng,

I'm pleased to inform you that your manuscript has been deemed suitable for publication in PLOS One. Congratulations! Your manuscript is now being handed over to our production team.

Kind regards,

on behalf of

Dr. Mingjun Hu

Academic Editor

PLOS One